# Evolving Symbolic 3D Visual Grounder with Weakly Supervised Reflection

## Abstract

Understanding the behavior of an end-to-end 3D visual grounder is challenging, especially when the grounder makes an unexpected prediction. Despite the llm-agent-based grounders performing step-by-step interpretable reasoning, the cost for evaluation at scale is prohibitive. To address the challenges, in this work, we propose a novel fully interpretable symbolic framework for 3D visual grounding, namely Evolvable Symbolic Visual Grounder (EaSe), with much less inference cost and superior performance. Given a symbolic expression of a grounding description translated by an LLM, EaSe calculates the feature of each concept utilizing a set of explicit programs in Python learned from a tiny subset of the training data. To learn this program library, we introduce a learning paradigm that continuously optimizes the programs on the training dataset by an LLM-based optimizer. We demonstrate that our paradigm is scalable when more data is involved. Experiments on ReferIt3D show EaSe achieves 50.7% accuracy on Nr3D, which surpasses most training-free methods and has considerable advantages in inference time and cost. On Sr3D, EaSe also has comparable overall performance with these approaches. Moreover, we perform extensive experiments to analyze the interpretability and feature quality and reveal the potential for reasoning and condition level grounding.

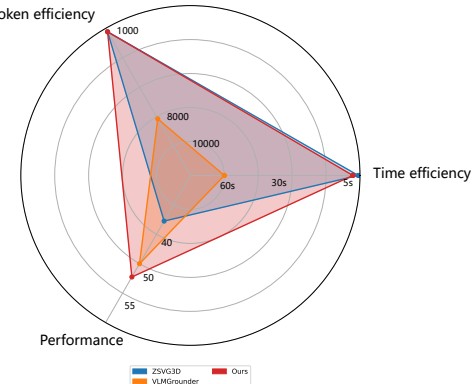

Figure 1: Comparison of EaSe with the two previous methods. With the symbolic framework and evolutionary self-refinement, EaSe excels in both performance and inference efficiency.

## 1 Introduction

The 3D visual grounding (3DVG) task aims to ground an object in a 3D scene based on a natural language utterance. There have been a lot of supervised methods for 3DVG(Achlioptas et al., 2020; Jain et al., 2022; Huang et al., 2022). By modeling various object attributes and spatial relations, and leveraging large-scale training data with high-quality annotations, these methods achieve high performances on 3DVG. These approaches are trained to have good performance in object detection, classification, attribute, and relation recognition. However, annotation of training data can be

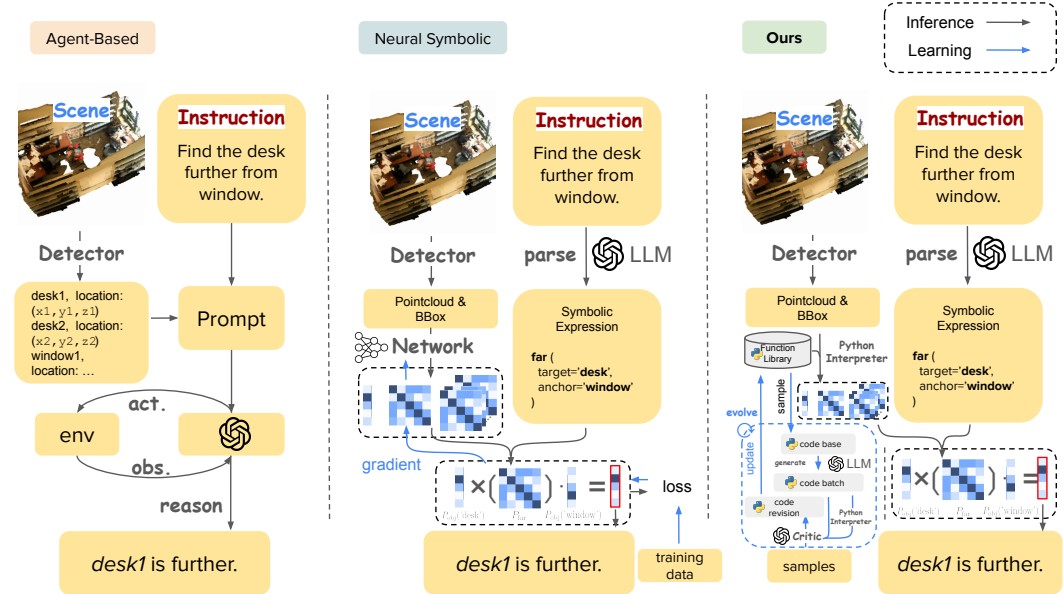

Figure 2: EASE uses LLM and training data to generate and refine the code for representing relations. During the evaluation, the executor can explicitly infer the target object.

expensive, and the limited vocabularies in the training data may limit the generalization and open vocabulary application in the real world.

Neuro-symbolic approaches (Hsu et al., 2023; Yuan et al., 2024) separate the processes of relation encoding and inference. In these methods, natural language descriptions are transformed into symbolic representations containing relevant categories and relationships, which are then encoded through a series of modules. The resulting expressions are executed using features to achieve grounding. However, these encoders either operate implicitly or rely on human annotation.

Subsequently, the expression is executed using features for grounding results. However, the encoders either are implicit(Hsu et al., 2023) or rely on human annotation(Yuan et al., 2024).

Recently, large language models (LLMs) have demonstrated significant capabilities in reasoning and generating executable code. Approaches that utilize LLMs and vision-language models (VLMs) simulate human-like grounding processes through multi-turn reasoning. These methods, leveraging the rich knowledge within pretrained models, are often training-free and support open-vocabulary applications. However, the requirement for multiple inferences to evaluate a single example leads to high computational costs and inefficiencies.

To provide explainable relation encoding and enable faster, more cost-effective inference, we introduce EASE, an evolvable 3D visual grounder that employs a symbolic framework and utilizes LLM generated Python codes as explainable encoders and pre-trained classifier to compute both relation and category features.

To improve the quality of generated codes, we designed a system that can automatically generate unit tests and give feedback based on test results. Then LLM can improve it through self-refine (Madaan et al., 2024). Considering some relations are associated, we use dynamic in-context examples retrieved from codes that have been generated for the generation of new relations. The distinction between EASE and previous approaches is illustrated in Figure 2. Our contributions are summarized as follows:

- We introduce EASE, a symbolic 3DVG approach employing LLM to generate explicit relation encoders by self-refinement without any human knowledge.

- By evaluating on 3D visual grounding experiments, EASE achieves 50.7% accuracy on Nr3D, surpassing previous training-free methods and having considerable advantages on time and token efficiency.

- We show the interpretability of both explicit relation encoders and step-by-step grounding. Besides, the features have certain properties without any and they are provable because of our interpretability.

## 2 METHOD

### 2.1 PROBLEM STATEMENT

3D visual grounding tasks involve a scene, denoted as $\mathcal{S}$, represented by an RGB-colored point cloud containing $C$ points, where $\mathcal{S} \in \mathbb{R}^{C \times 6}$. Accompanying this is an utterance $\mathcal{U}$ that describes an object within the scene $\mathcal{S}$. The objective is to identify the location of the target object $\mathcal{T}$ in the form of a 3D bounding box. In the ReferIt3D dataset (Achlioptas et al., 2020), bounding boxes for all objects are provided, making the visual grounding process a task of matching these bounding boxes to the scene $\mathcal{S}$. In contrast, the ScanRefer dataset (Chen et al., 2020) provides only the scene point cloud, requiring additional detection or segmentation modules to accomplish the grounding task.

### 2.2 GROUNDING PIPELINE

We adhere to the previous SOTA neuro-symbolic framework for 3DVG (Hsu et al., 2023; Feng et al., 2024). The grounding pipeline is composed of three main components: the semantic parser that converts $\mathcal{U}$ into a structured expression $\mathcal{E}$; encoders to compute the features for descriptive terms such as `near` and `small` within $\mathcal{E}$. Subsequently, an executor performs logical reasoning over $\mathcal{E}$ and computes matching scores between $\mathcal{S}$ and each object.

**Semantic parser.** We employ GPT-4o (OpenAI, 2024) as the semantic parser and structure the expressions in JSON format, which consists of the following components:

- **category**: A string representing the category of the target object referenced in $\mathcal{U}$.
- **relations**: A list defining the spatial constraints relative to the target object. Each entry in this list includes:
  **relation_name**, a string specifying the spatial relation mentioned in $\mathcal{U}$, such as "near" or "above."; **objects**, a list of objects that share the specified relation with the target object. Each element is represented as a separate JSON object; **negative**, A boolean indicating that if set to true, the target object should not exhibit this relation.

For example, the phrase "chair near the table and under a shelf" can be represented as:

```
{"category": "chair", "relations": [{"relation_name": "near",
"objects": [{"category": "table"}]}, "relation_name": "below",
"objects": [{"category": "shelf"}]}]}
```

Human-annotated natural language expressions exhibit diverse descriptions of relations, leading to a long-tail distribution of **relation_name** in parsed expressions. To address this issue, we pre-define a set of common relation names and prompt LLM to select from them for $\mathcal{E}$ instead of using the original word from $\mathcal{U}$. Based on the number of associated objects, the relations are categorized into `unary`, `binary`, and `ternary` (Feng et al., 2024). Table 1 presents our predefined set of relations along with their classifications. For simplicity, attributes that describe properties of a single object, such as "large" or "at the corner," are treated as special types of relations.

**Relation encoder.** Neuro-symbolic approaches always use neural network-based encoders to compute features corresponding to category names and relation names in $\mathcal{E}$ for grounding. Unlike previous neuro-symbolic approaches, our relation encoder is a Python class generated by LLM. It directly utilizes the scene's point cloud data, performing computations in a transparent and interpretable manner. Additionally, object classification is conducted using a pre-trained point cloud classifier.

Table 1: Classification of all relations.

| Classification | Relations |
|---|---|
| unary | large, small, high, low, on the floor, against the wall, at the corner |
| binary | near, far, above, below, left, right, front, behind |
| ternary | between |

The feature of categories and unary relations, denoted as $f_{\text{unary}} \in \mathbb{R}^N$, can be seen as the matching score between objects and their respective categories or relations, where $N$ is the number of objects in the scene. Features of binary relations $f_{\text{binary}} \in \mathbb{R}^{N \times N}$ represent the likelihood of binary relations existing between all possible pairs of objects. For instance, the element $f_{\text{near}i,j}$ quantifies the probability that the $i$-th object is "near" the $j$-th object. The structure of ternary features follows a similar pattern.

**Executor.** The executor use $\mathcal{E}$ and associated features to identify $\mathcal{T}$. Since elements in the features represent the probabilities or corresponding relation or category, the logical conjunction in $\mathcal{E}$ can be represented through the product operation. For a symbolic expression $\mathcal{E}$, the classification score $c_{\text{cls}}$ of its `category` is calculated initially by a classifier. Subsequently, the executor processes each relation individually by referring to the `relations` field. For every relation, the relation feature $f_{\text{relation}}$ is computed, and the grounding scores $score_{\text{sub}} \in \mathbb{R}^N$ for the corresponding objects can be recursively calculated. Since the number of associated objects is always less than the feature dimension by one, the grounding result of the individual relation can be determined through a dot product between feature and grounding scores. Once all grounding results have been computed, the final score $score \in \mathbb{R}^N$ is obtained by performing an element-wise (Hadamard) product between $c_{\text{cls}}$ and all $score_{\text{relation}}$. The target object's index is then determined by applying the argmax operation to the final score.

## 2.3 RELATION ENCODER

The sizes and positions of objects in 3D scenes are mathematically related to specific relations. For example, "near" is related to the distance between objects, and "large" is related to the volume of an object. In EASE, we represent each relationship in a modular format using Python classes. Each class contains a highly interpretable computing process. These classes are generated by LLM through generation and self-refinement (Madaan et al., 2024) on a little scale of data from the training set. Previous works Yuan et al. (2024); Fang et al. (2024) also employ functions for relation computation. However, a key distinction to EASE is that it eliminates the need for human annotations or specific prompts—either at the code or text level—thereby reducing reliance on human knowledge. 2.3.1 outlines the overarching structure of our classes, while the data collection process is detailed in 2.3.2. In 2.3.4, we describe the procedures for generating and refining the classes. The overall framework is depicted in Figure 3.

### 2.3.1 CLASS STRUCTURE

For each relation, we define a corresponding Python class. The class is initialized with the point cloud data of the scene, represented as a PyTorch tensor, and has two primary methods. One is `_init_param`, which is used to compute all parameters for derive features such as the distances between pairs of objects in the case of the "near" class. The other is `forward` executes numerical computations, including operations such as inversion and exponentiation, and ultimately returns the computed feature.

### 2.3.2 UNIT TEST

LLMs may not always generate a perfect code within one attempt(Olausson et al., 2023). Therefore, we incorporate relation-specific data from the training set to enhance both accuracy and quality, enabling the LLM to refine its code based on these data samples.

The symbolic framework facilitates the collection of this data. For each relation, we use the heuristics method to filter the training set data based on the parsed expressions, excluding samples with simple structures. These data typically include only object categories and relationship labels, lacking more complex logical constructs. The data scale for each relation is small (less than 100).

This approach allows the use of straightforward numerical relationships for testing. We write a test suite to compute the relation feature using generated code and compare the corresponding value of the target object with that of distractors from the same class. The code passes the test case if the target object's corresponding value is the highest. The test suite also provides detailed feedback on failed cases. For example, in the case of the relation "small", the target object, being smaller, should have a larger corresponding feature value compared to distractors. If a distractor exhibits a larger value, this test case fails, and bounding box information of both the distractor and target object will be given for the feedback message in 2.3.4.

### 2.3.3 PROMPT

Our code generation prompt is primarily composed of two sections: the general instruction and the in-context example.

**General Instruction**   The general instruction provides fundamental details, such as the task description, execution environment, and class schema. To ensure more precise code generation for view-dependent relationships, we include additional specific guidelines. Please refer to the appendix for the prompt.

**In Context Example**   In-context examples can assist LLM generate more accurate responses (Brown et al., 2020). However, fixed annotated in-context examples may not work for all relations. On the other hand, annotating specific examples for every relation needs a lot of human effort and could lead to the model relying more on the explicit human-provided knowledge within the examples than on the knowledge encoded in the model itself. Some relations have exhibited computational similarities. For example, code for computing relation "near" and "far" may be largely identical in computing pair-wise distances but differ only in the numerical processing. Once a well-refined code for "near" is developed, it can serve as an in-context example for generating responses related to "far". By employing dynamic in-context examples, this method offers two key benefits: 1) reduced reliance on extensive human annotation compared to manually annotated in-context examples (ICE), and (2) enhanced efficiency and precision in generating similar relational data when compared to settings where no ICE is available.

We prompt LLM with instruction `select the relations that may be relative to ...   from following` to construct a directed acyclic graph $\mathcal{G}$ to implement that. In $\mathcal{G}$, each node represents a relation, and an edge from node A to node B means that when generating code for B, code of A is used as ICE. The entire code generation process can then follow a topological sort order. We show the graph in Figure 7.

### 2.3.4 CODE REFINEMENT

The code generation and refinement are done in many iterations. In the initial iteration, we give the prompt in 2.3.3 and relation name to LLM to generate $N_{sample}$ responses. where $N_{sample}$ is a hyperparameter. Following this unit tests in 2.3.2 are executed on all generated codes. If a code can pass all the test cases, we use it as the final choice and stop the generation. Otherwise, we randomly select 3 failure cases from the test suite 2.3.2 and formulate a feedback message for refinement. This feedback specifies the expected execution outcomes and includes bounding box information from the test suites, instructing the LLM to modify the code accordingly.

In subsequent iterations, codes having $top\_k$ highest pass rate on test cases in the last iteration are selected for further refinement. For each code, its feedback message is appended and the LLM generates another $N_{sample}$ modified version based on the old version and the feedback message. The same testing and refinement process is applied to these new samples. After $N_{iter}$ iterations, the code with the highest pass rate is chosen as the final version. If multiple samples achieve the same pass rate, we select from which is refined more times.

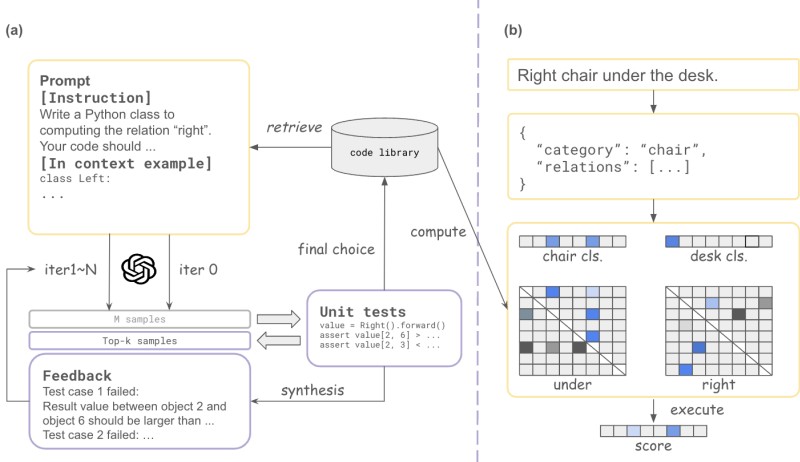

Figure 3: Overview of our framework. (a): the process of code generation and refinement. We retrieve an ICE for the first iteration and get the refined code by filtering and self refinement for many iterations. The final version is stored in a library for other relations. (b): In testing time, features are computed by pretrained classifiers and generated codes. They are executed on the parsed symbolic expression for the target object.

## 3 EXPERIMENTS

### 3.1 EXPERIMENTAL SETTINGS

**Dataset**   We conduct experiments on ReferIt3D (Achlioptas et al., 2020) dataset, which has 2 subsets: Nr3D and Sr3D. Nr3D subset utterances contain human-annotated utterances and Sr3D contains synthesized ones. Based on the number of same-class distractors, the dataset can be categorized into "easy" and "hard" subsets depending on the number of same class distractors. The easy subset has a single distractor and the hard subset has multiple. The dataset can also be split into "view dependent" and "view independent" subsets according to if there are some keywords in the utterance. Ground truth object bounding boxes are given in ReferIt3D default test setting. So the metric is an exact match between the predicted bounding box and the target bounding box.

**Implementation Details**   In code generation, we set $N_{sample}$ and $N_{iter}$ to 5, $top_k$ is 3. We mainly use gpt-4o-2024-08-06 model with a temperature of 1.0 and top_p of 0.95. For a fair comparison, we use the same object classification result and evaluation code as Yuan et al. (2024).

**Baselines**   We compare EASE with supervised approach BUTD-DETR (Jain et al., 2022), neruosymbolic approaches NS3D (Hsu et al., 2023), ZSVG3D (Yuan et al., 2024) and agent based approaches Transcrib3D (Fang et al., 2024), VLMGrounder (Xu et al.).

We compare EASE with them on performance, inference cost on ReferIt3D (Achlioptas et al., 2020) and visualize some grounding examples and features to assess the quality.

### 3.2 QUANTITATIVE RESULTS

**ReferIt3D**   As Shown in Table 2, in settings where ground truth labels are unavailable, the overall performance of EASE outperforms the zero shot baseline ZSVG3D (Yuan et al., 2024) and VLM-Grounder (Xu et al.). When compared to the previous state-of-the-art supervised method, BUTD-DETR, EASE achieves comparable performance on view dependent subset but still lags behind on view independent subset.

To compare with Transcrib3D (Fang et al., 2024) fairly, we utilize GT labels for evaluation. In this setting, EASE has a comparable performance on view dependent subset but the gap on view

Table 2: Performance on Nr3D, VD, and VID stands for view-dependent and view-independent, respectively. Performance of EASE is comparable with previous zero shot methods and surpasses the baseline ZSVG3D by 11.4%. Meanwhile, EASE offers significant advantages in terms of time efficiency and token costs. †: VLM-Grounder is evaluated on a subset having 250 samples. * : we re-run ZSVG3D using gpt-4o.

| Method | Overall | Easy | Hard | VD | VID | Time | Token |
|---|---|---|---|---|---|---|---|
| BUTD-DETR (Jain et al., 2022) | 54.6 | 60.7 | 48.4 | 46.0 | 58.0 | - | - |
| ZSVG3D* (Yuan et al., 2024) | 39.3 | 45.5 | 33.0 | 38.2 | 40.0 | 0.5 | 305 |
| VLM-Grounder† (Xu et al.) | 48.0 | 55.2 | 39.5 | 45.8 | 49.4 | 60.0 | 8000 |
| EASE (Ours) | **50.7** | 58.7 | 43.0 | 45.6 | 53.2 | 2.1 | 1178 |
| Transcrib3D (Fang et al., 2024) | 70.2 | 79.7 | 60.3 | 60.1 | 75.4 | 27.0 | 12215 |
| EASE (Ours, w/ GT Labels) | 65.7 | 75.6 | 56.2 | 58.7 | 69.1 | 2.1 | 1178 |

Table 3: Performance on Sr3D and the Nr3D subset in NS3D.

| Method | Overall | NS3D |
|---|---|---|
| BUTD-DETR (Jain et al., 2022) | 67.0 | - |
| NS3D (Hsu et al., 2023) | 62.7 | 52.6 |
| NS3D (Hsu et al., 2023) (w/ GT Labels) | 96.9 | - |
| Transcrib3D (Fang et al., 2024) (w/ GT Labels) | **98.4** | - |
| EASE (Ours) | 62.1 | **59.9** |
| EASE (Ours, w/ GT Labels) | 95.3 | - |

independent subset remains significant. Nevertheless, EASE exhibits superior efficiency in terms of both time and token usage.

We also evaluate the performance on Sr3D and the Nr3D subset proposed by NS3D (Hsu et al., 2023). Shown in Table 3, EASE's overall performance is close to other baseline methods on Sr3D. And on the NS3D subset, EASE surpasses the baseline NS3D by 7.3%.

**Time and cost evaluation** Both VLM-Grounder (Xu et al.) and Transcrib3D (Fang et al., 2024) are agent-based methods, which result in high computational time and token usage during inference due to multiple LLM/VLM calls. We evaluated Transcrib3D and VLM-Grounder on a randomly selected set of 50 examples from the Nr3D dataset, with the average time and token costs during inference displayed in the two rightmost columns of Table 2.

In contrast, EASE demonstrates lower time and token costs. VLM-Grounder can require up to 60 seconds and 8000 tokens. Although Transcrib3D has a slightly shorter average inference time, it remains significantly longer than EASE and incurs a substantial token cost of over 12,000. The resource efficiency of EASE is comparable to ZSVG3D, which employs a similar framework and achieves slightly lower costs due to batch prompting. Compared to agent-based methods, EASE reduces more than 90 % inference time and token consumption.

### 3.3 QUALITATIVE RESULTS

**Scene Visualization** In Figure 4, we show two step-by-step grounding cases of EASE illustrating how the final grounding results are constructed through a series of smaller, intermediate grounding steps.

The corresponding utterance of the second row is "When facing the door, it's the shelf above the desk on the right." Execution of the parsed symbolic expression can be seen as 4 steps, progressing from left to right. Each of the 4 subfigures shows the internal grounding results and the corresponding utterance. Objects with higher scores are highlighted in green, with lighter objects showing a better match to the utterance.

In the leftmost subfigure, the phrase "right to the door" is grounded by computing the dot product between the feature representing the "right" spatial relation and the classification score for "door."

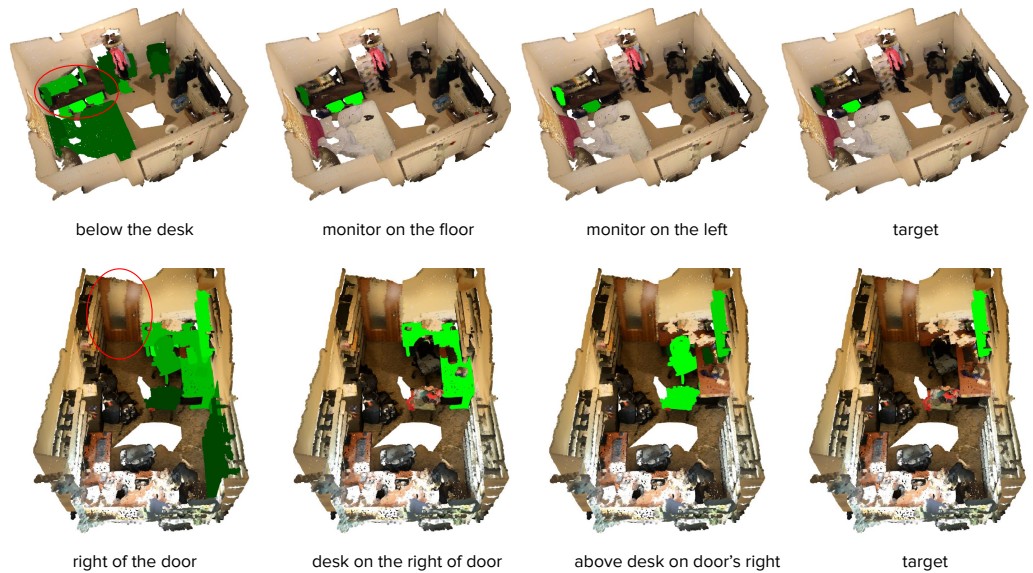

below the desk      monitor on the floor      monitor on the left      target

right of the door      desk on the right of door      above desk on door's right      target

Figure 4: Explanation and visualization of the grounding steps. Anchors (desk for the upper case and door for the lower) are marked by **red circle**. We only visualize a part of objects that match the below conditions well in **green**, objects of brighter color have higher scores of meeting the condition.

The grounding result for "desk on the right of the door" is obtained by applying the Hadamard product to the previous result and the classification score for "desk." Through similar operations, the entire utterance is grounded, making the process highly interpretable.

**Relation constraints**   Feng et al. (2024) propose that some spatial relations are symmetric, like "near" or "far", which means if object A is "near" B, B should be also "near" A. So the features of these relations should be symmetric. Some other relations should be asymmetric like "left" or "right". In these cases, if a feature element is positive (indicating the presence of the relationship), its corresponding symmetric element should be zero (indicating the absence of the reverse relationship). Feng et al. (2024) add loss for these constraints during training for regularization. Even if there is not any training or special instruction from humans on them, we observe that in our generated data, the constraints can be ensured on relation "near", "far", "left" and "right" because of the deterministic execution of code. This means the feature of 'near' and "far" is guaranteed to be symmetric. For features $f$ for "left" and "right", if $f_{i,j} > 0$, $f_{j,i}$ is guaranteed to be 0.

**Error Breakdown**   We randomly selected 100 error cases from EASE on the Nr3D to conduct a detailed error analysis. Results are shown in section 3.3. The primary failure source is related to errors in feature computation. These errors come from flaws in codes or inaccuracies in object classification. Another significant source of failure is the limitations of our system design. Specifically, we simplify the scene representation into a list of 3D bounding boxes with predicted labels, while omitting critical details such as object orientation, shape, color, and other visual attributes. Furthermore, the system does not incorporate region or room-level segmentation of the scenes. Additionally, some failure cases are linked to issues in the semantic parsing process. On the one hand, we only have a limited set of common relations, which is insufficient for grounding utterances in real-world scenarios. On the other hand, LLM cannot ensure the generation of accurate symbolic expressions that align with the grounding utterance, which can result in incorrect grounding results.

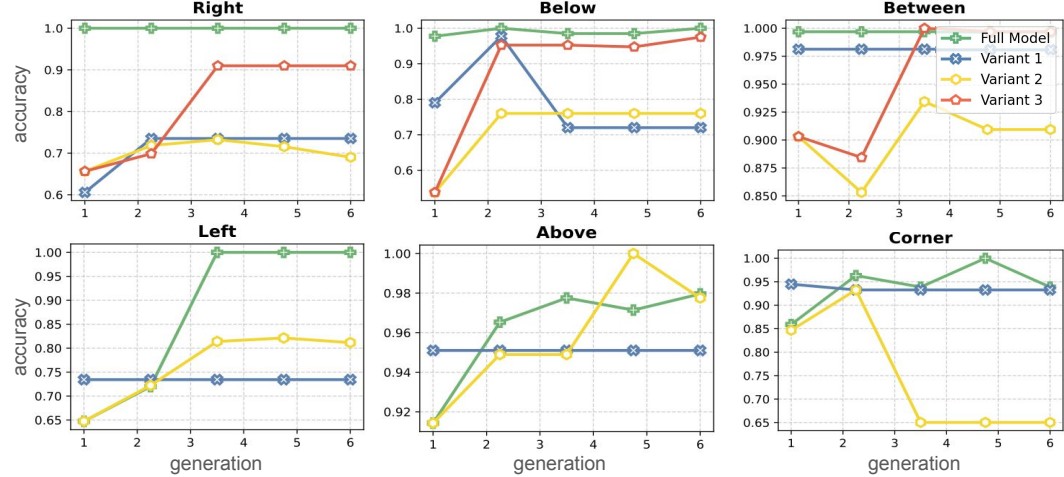

Figure 6: The learning curves of different variants. The x-axis is the generation number of the program. The y-axis is the accuracy of the program on the test set. It is worth noting that the curves of our full model ──╬── coincide with the curves of Variant 3 ──○── in the second row ("**Left**", "**Above**", and "**Corner**") since Variant 3 can be regarded as the first stage of generating all programs.

## 3.4 ABLATION STUDIES

In this section, we conduct an ablation study to investigate the impact of various components during the code generation process, evaluating three different variants. In all three variants, no ICE2.3.3 is provided. 1) Direct Code Generation: In this variant, we generate the code by prompting the LLM with the prompt for initial iteration, and sample multiple codes. The code with the highest pass rate on unit tests is selected. 2) General Refinement: In this variant, we use the same prompt as variant 1 and replace the feedback message in 2.3.4 with a general refinement instruction. Please refer to the appendix for the details. 3) In-context Example Ablation: In this variant, we only ablate the in-context examples. For the relations without in-context examples in the main experiment, variant 3 is identical to the first stage of our full model, so we only plot variants 1 and 2 for "left", "above", and "corner". For relations "right", "below" and "between", we conduct experiment on

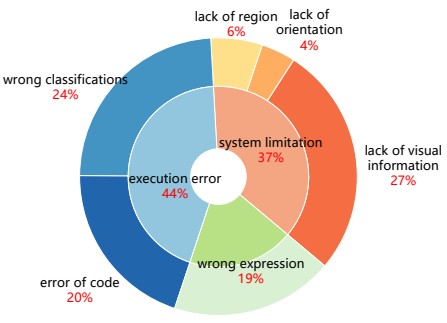

Figure 5: Error breakdown result.

all 3 variants. To control for the impact of the first iteration, we use the same responses in iteration 0 across variant 2 and variant 3.

Figure 6 shows the results of the ablation study, different variants are represented by lines of different colors. The horizontal axis represents the number of iterations, after an iteration ends, we choose a code based on their unit test pass rates and then evaluate it on the test set. The vertical axis shows the number of correctly solved test examples associated with the relation. We normalize them by dividing them by the maximum value. We can see that random sampling (Variant 1) shows comparable performance to the full model only for the "corner" and "between" relations. In Variant 2, which uses simple self-reflection, there is noticeable performance improvement across most relations except "corner." This variant achieves performance on "above" and "corner" close to that of the full model, as these relations are easier to generate. However, it struggles with the more complex relations. Variant 3, which incorporates feedback messages, improves upon the performance of Variant 2. For all relations except "right", it achieves results similar to the full model. However,

for "right," "between," and "below" — where in-context examples are used to assist generation — Variant 3 shows a significant performance gap in the early iterations compared to the full model, highlighting the impact of in-context examples.

## 4 RELATED WORK

**3D Visual Grounding** 3D visual grounding focuses on finding an object in 3D scene based on descriptions to appearance or location. ScanRefer (Chen et al., 2020) and ReferIt3D (Achlioptas et al., 2020) are 2 popular benchmarks on this task, providing rich and diverse object-utterance pairs on ScanNet (Dai et al., 2017). Traditional approaches train a end-to-end model to work on this task. By encoding more features on visual information (Huang et al., 2022), designing fine-grained encoder on spatial relationship or training on large scale well-annotated data (Ziyu et al., 2023). Recent approaches employ large multi-modal models for reducing training data, enhancing perception and reasoning. Neural symbolic approaches (Hsu et al., 2023; Feng et al., 2024; Yuan et al., 2024) use LLM to parse natural language grounding utterances into executable structured expressions, improving data efficiency. R2G (Li et al., 2024) employ scene graph to modeling object attributes and relations, It surpasses previous methods on view dependent utterances. Agent-based approaches (Yang et al., 2024; Fang et al., 2024) create virtual environments where LLM can take actions, get observations and do iterative reasoning. Xu et al. use VLM and images from the scene to figure out the target object without the need for detection or segmentation modules.

**LLM Programming** LLM can generate executable code (Roziere et al., 2023; Zhu et al., 2024). Many works use LLM's programming ability on reasoning (Li et al., 2023a), robotic controlling (Liang et al., 2023), reward designing (Xie et al.; Ma et al., 2023). For high-quality and stable code output, Le et al. (2022); Chen et al. (2023) use the feedback from the outside environment to fine-tuning LLM, Ma et al. (2023) use RL training trajectories to select and improve functions.

**Neural Symbolic Reasoning** Neural symbolic reasoning methods parse natural language to symbolic expression and get reasoning results by executing expressions. Li et al. (2023b) provides a neuro-symbolic interface that supports many logical reasoning and seamless integration with other pretrained model seamlessly. Cheng et al. (2023) use LLM query to assist SQL execution. Mao et al. (2019); Hsu et al. (2023) define domain-specific language and use neural networks to encode labels and relations. Zero-shot visual programming (Gupta & Kembhavi, 2022; Yuan et al., 2024) use Python code as the symbolic language and execute by interpreter equipped with predefined APIs.

## 5 CONCLUSION & LIMITATION

In this work, we propose a way to encode relations in Python programs for symbolic 3D visual grounding and a weakly supervised framework for filtering and refining LLM synthesized codes. We directly use knowledge distilled from LLM for relation encoding instead of human annotation or supervised learning. The whole system is highly explainable. We demonstrate its advantages in performance, data efficiency, and inference cost.

However, some limitations still remain. In natural language referring sentences, there are diverse descriptions of the relation. But we constraints the domain of keywords in a prompt and pass a minor part of spatial relation names during semantic parsing. Besides, we treat the scene as a list of 3D bounding boxes and ignore the object's appearance, orientation, and room-level information of the scene.

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

# A PROMPTS

## A.1 SEMANTIC PARSING

In this section, we show the prompt for semantic parsing. The relation set in the prompt is slightly larger than that in Table 1 and we create a lookup table for replace relations in parsed expressions with relations in Table 1.

### Listing 1: prompt for semantic parsing

```
You are a skilled assistant with expertise in semantic parsing.

## Task Overview
I will provide you with a sentence that describes the location of an object within a scene.
    Your task is to convert this description into a JSON format that captures the essential
    details of the object.

### The JSON object should include:
- **"category"**: string, representing the object's category.
- **"relations"**: a list of relationships between the object and other elements in the scene.
     Each relationship should be represented as a dictionary with the following fields:
    - **"relation_name"**: string, specifying the type of relationship. The relationship can
         be:
        - *Unary*: choose from ['corner', 'on the floor', 'against wall', 'smaller', 'larger',
             'taller', 'lower', 'within'].
        - *Binary*: choose from ['above', 'below', 'beside', 'close', 'far', 'left', 'right',
             'front', 'behind', 'across'].
        - *Ternary*: choose from ['between', 'center', 'middle'].
        Only consider **simple** and **general** relations, donot make complex ones like "left
             of a blue box", "with dark appearance", "facing the window", etc. You should
             handle these by logical structures.
        If the relationship is not mentioned in the list, you should choose the most
             appropriate relation above. **Never** create a new relation name!
    - **"objects"**: a list of objects involved in the relationship. Every object in the list
         should have the same JSON structure. The list structure depends on the relationship
         type:
        - *Unary*: The list should be empty.
        - *Binary*: The list should contain one object.
        - *Ternary*: The list should contain two objects.
    - **"negative"**: boolean, indicating if the object is explicitly described as not having
         this relationship. Set this to True if applicable.

## Guidelines:
- First, generate a plan outlining the object's appearance and relationships based on the
     sentence. Then, use this plan to create the JSON representation.

## Examples:

### Example 1:
**Sentence**: The correct whiteboard is the one on a table.
**Plan**: "Correct" does not describe appearance. The appearances are "whiteboard" and "table
    ", and the "whiteboard" is on the "table".
**Parsed JSON**:
```json
{
    "category": "whiteboard",
    "relations": [
        {
            "relation_name": "above",
            "objects": [
                {
                    "category": "table",
                    "relations": []
                }
            ]
        }
    ]
}
```

... 2 more examples.
```

### Listing 2: prompt for generation for "right"

```
You are an expert on spatial relation analysis and code generation.

# Introduction to task
```

702

703 Your task is to write a Python class which can be used to compute the metric value of the
704     existence of some spatial relationship between two objects in a 3D scene given their
        positions and sizes. Higher the metric value, higher the probability of the two objects
705     have that relation.

706 In the class, you will be given the positions and sizes of the objects in the scene. The class
707     should have a method `forward` which returns a tensor of shape (N, N), where element (i,
        j) is the metric value of the relation between object i and object j.
708

709 In the 3D scene, x-y plane is the horizontal plane, z-axis is the vertical axis.

710 # Introduction to programming environment

711
712 Here is an example class for `Left` relation. The class you write should have the same
        structure as the example class.

713
714 ```python
    class Left:
715     # ...
    ```
716 Make sure all tensors are placed on `DEVICE`, which has been defined in the environment.
717 The code output should be formatted as a python code string: "```python ... ```".

718 # Some helpful tips

719
720 (1) You should only use the given variables, and you should not introduce new variables.
    (2) The metric value should be sensitive to the input arguments, which means if the arguments
721     change a little, the value should change a lot.
722 (3) The metric value should be 0 if the two objects don't have that relation, never set
        negative values!
723 (4) Never treat an object as its center point, you must consider the size of the bounding box,
        just like the example code. Never set an threshold to determine the relation. The value
724     of the relation should be continuous and sparse.
    (5) You should imagine that you are at position (0, 0) to determine the relative positions.
725 (6) Remember you are **in** the scene and look around, not look from the top. So never use the
726     same way as 2D environment.
    ...
727

728 Propose your method first and then generate the code. Think step by step.
729 Don't use any axis or specific direction as the reference direction or right direction, your
        method should work for any perspectives.
730

731

732                         Listing 3: an example for feedback message

733 We have run your code on some cases. Here are 3 failure cases:

734 # Case 1.

735
736 Metric value of object tensor([ 0.3992, -0.5619,  0.8831,  0.3921,  0.3476,  0.1059], device='
        mps:0') "above" object tensor([-0.0432, -0.6965,  0.8483,  0.6526,  0.4943,  0.3061],
737     device='mps:0') should be larger than 0. Metric value of object tensor([ 0.3992, -0.5619,
         0.8831,  0.3921,  0.3476,  0.1059], device='mps:0') "above" object tensor([-0.0432,
738     -0.6965,  0.8483,  0.6526,  0.4943,  0.3061], device='mps:0') should be higher than the
        metric value of object tensor([0.5338, 1.1607, 1.1160, 0.2121, 0.3323, 0.8192], device='
739     mps:0') "above" object tensor([-0.0432, -0.6965,  0.8483,  0.6526,  0.4943,  0.3061],
740     device='mps:0'). # Case 2.

741
742 Metric value of object tensor([-0.3941,  1.5280,  0.4675,  0.5132,  0.3938,  0.1191], device='
        mps:0') "above" object tensor([-0.4468,  2.1906,  0.5153,  0.9604,  2.0905,  0.9733],
743     device='mps:0') should be larger than 0. Metric value of object tensor([-0.3941,  1.5280,
         0.4675,  0.5132,  0.3938,  0.1191], device='mps:0') "above" object tensor([-0.4468,
744     2.1906,  0.5153,  0.9604,  2.0905,  0.9733], device='mps:0') should be higher than the
        metric value of object tensor([-0.0855,  3.4164,  0.2426,  0.4462,  0.5911,  0.1475],
745     device='mps:0') "above" object tensor([-0.4468,  2.1906,  0.5153,  0.9604,  2.0905,
746     0.9733], device='mps:0'). # Case 3.

747
748 Metric value of object tensor([-2.1831,  2.9738,  1.0996,  0.3647,  0.2885,  0.5714], device='
        mps:0') "above" object tensor([-1.9380,  2.3704,  0.5013,  0.7656,  1.2784,  0.8153],
749     device='mps:0') should be larger than 0. Metric value of object tensor([-2.1831,  2.9738,
         1.0996,  0.3647,  0.2885,  0.5714], device='mps:0') "above" object tensor([-1.9380,
750     2.3704,  0.5013,  0.7656,  1.2784,  0.8153], device='mps:0') should be higher than the
        metric value of object tensor([1.4070, 3.5247, 0.7835, 0.3882, 0.3539, 0.5908], device='
751     mps:0') "above" object tensor([-1.9380,  2.3704,  0.5013,  0.7656,  1.2784,  0.8153],
752     device='mps:0').

753
754 The first three are the center of the object, the last three are the size of the object. x-y
        is the horizontal plane and z is the vertical axis.
    After test, the pass rate of your code is too low. So you MUST check carefully where the
755     problem is. If you can't find the problem, you should
    come up with a new algorithm and re-write your code.

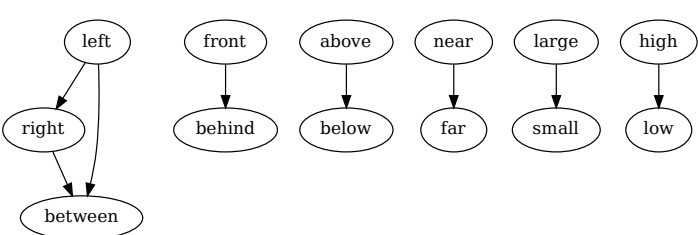

Figure 7: The graph for in context example selection.

```
Don't forget the following tips:
(1) You should imagine that you are at position (0, 0, 0) to determine the relative positions.
(2) Remember you are **in** the scene and look around, not look from the top. So never use the
       same way as 2D environment.
(3) Don't use any of x-axis or y-axis as your perspective, Your method should work for every
       perspective.
(4) The horizontal plane is x-y plane.
Please carefully analyze each of the failure case and explain why your code failed to pass it.
       The reason can be incorrect test case might or your code might not be able to handle
       some specific cases. Please write your analysis for each of the failure cases.

After the analysis of all cases, you should write the improved code based on your analysis.
       But **never** modify on the class methods and function parameters.

Some possible improvement ways:
1. Use a new algorithm to calculate the metric value rather than just modifying the existing
       code.
2. Consider carefully what other factors might be relevant to the spatial relationship between
       two objects and use them in your calculation.
3. Check the correctness of the input data and the calculation process.
```

## A.2 SELF-REFLECTION PROMPT

We show self-reflection prompt used in variant2, subsection 3.4 here.

Listing 4: prompt for self-reflection

```
Reflect on the code above, think carefully how to make it better. For example, check if you
       ignore some factors that may affect the result or use a wrong method.
Then you must re-write the code in the same format. Remeber all the tips!
```

## B EXECUTION DETAILS

In this section, we introduce the detailed algorithm of our execution.

**relation lookup table**  Some relations are not in the pre-defined set in 2.3.3, but they have similar meaning with some relations in the set. So we build an lookup table from relation names in the set to parsed relation names for relation translation.

Listing 5: lookup table

```
"near": ["near", "beside", "next to", "within", "in", "inside", "close", "closer", "closest",
       "surrounded by", "around", "facing", "with", "attached"],
"far": ["far", "farthest", "opposite", "furthest", "cross", "across"],
"corner": ["corner"],
"against wall": ["against wall"],
"above": ["above", "on top", "on the top", "on"],
"below": ["below", "under", "beneath"],
"tall": ["higher", "taller", "highest", "upper"],
"low": ["lower"],
"on the floor": ["on the floor"],
"small": ["smaller", "shorter"],
"large": ["larger", "bigger", "largest", "longer"],
```

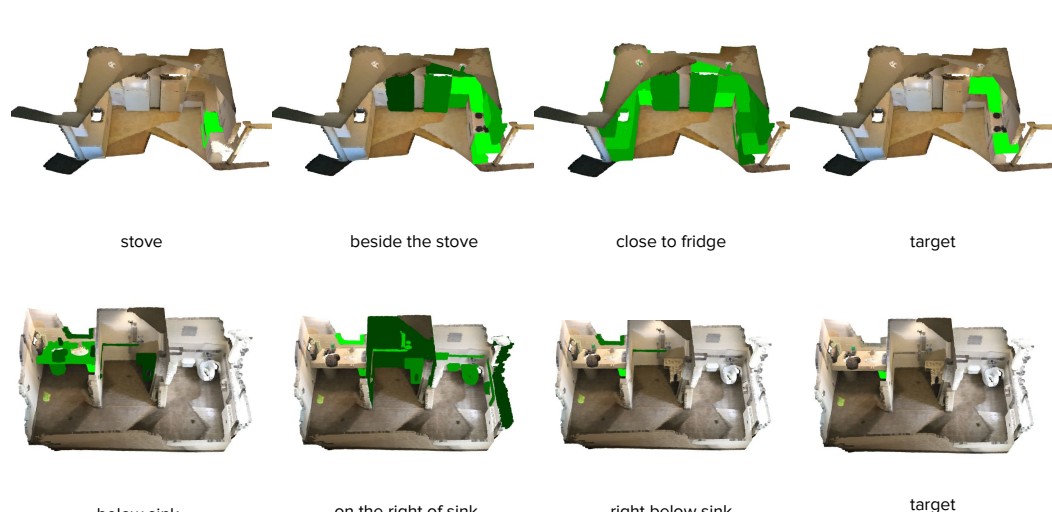

stove      beside the stove      close to fridge      target

below sink      on the right of sink      right below sink      target

Figure 8: More visualization results.

```
"left": ["left"],
"right": ["right"],
"front": ["front"],
"behind": ["behind"],
"between": ["middle", "center", "between"]
```

**category feature** Yuan et al. (2024) provides the classified label for each objects. To convert labels to an feature $f_{\text{category}} \in \mathbb{R}^N$, we calculate the text similarity between `category` and all labels $sim \in \mathbb{R}^N$ using CLIP(Radford et al., 2021). Then we use $f_{\text{category}} = softmax(100 \times sim)$ as the category feature.

**negative condition** If the field `negative` is true, the objects having higher value in the feature should be degraded. In the execution, we use $f_{\text{neg}} = max(f) - f$ to implement that.

## C  EXTRA EXPERIMENT RESULTS

### C.1  VISUALIZATION OF SCENES

In this section, we visualize 2 more grounding examples in Figure 8. The first row shows the process of grounding "the kitchen cabinet close to fridge and beside the stove"; the second row is the grounding of "trash can on right below the sink".

### C.2  CONDITION LEVEL GROUNDING

Our parsed symbolic expressions actually contain one or more spatial conditions of the target object. However, there may be some redundant condition in the utterance. Take the first row of Figure 4 for example, because all monitors "on the floor" are "under the desk", so one of these two conditions are redundant, which means even if the method can not process one condition of them, it can still give the correct grounding result. So we test EASE on utterances containing single condition for a better understanding of its ability.

We categorize objects of same class to groups. With in a group, we collect the conditions for each object from parsed expressions. Each condition is a JSON format like {{"relation": ..., "objects": [...]}} and can be executed seamlessly to find best matching object.

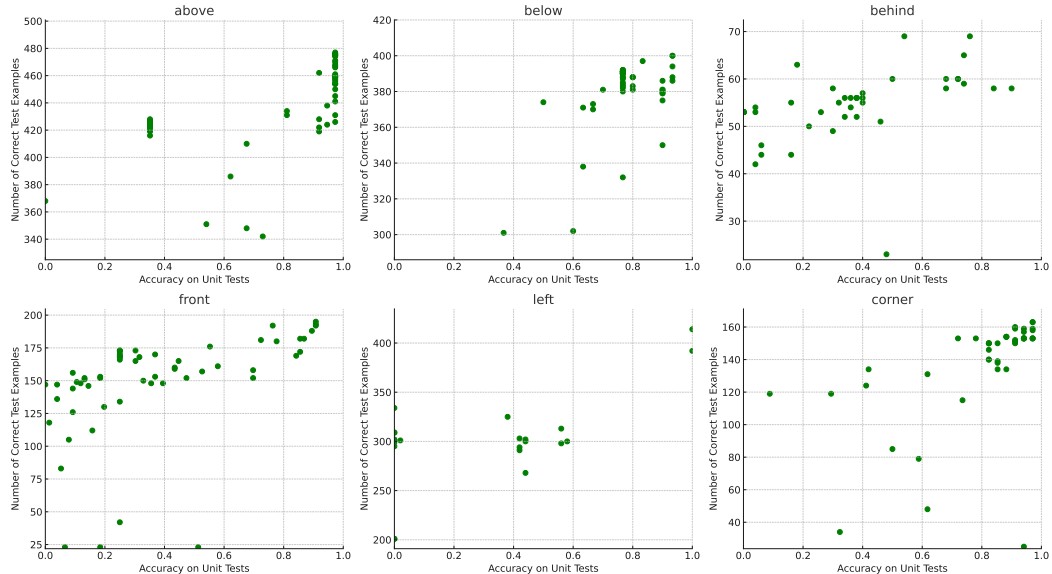

Figure 9: Corresponding relation between the unit test pass rate and number of correct examples on test set.

We calculate the average precision and recall for all condition level matches. EASE has 67.5% average precision and 66.9% average recall.

## C.3 EFFECT OF UNIT TESTS

To demonstrate the effect of filtering generated code by its accuracy on training cases, we choose 6 relations and plot the pass rate on training cases on the x-axis and the number of passed examples in all relative test examples. For some easy relations like "near" or "far", GPT-4o can pass all the tests at once, so we only show the cases having multiple refinement steps.

The result is shown in Figure 9. For 5 of 6 relations (except relation behind), the code having the highest performance on the training cases can have the top-tier performance on the test set. As for relation behind, using the best code on training cases causes about 15 cases loss on the test case compared to using about 70 percent accurate code. But it's still better than using most of the codes whose accuracy is less than 0.5. This might be caused by the bias of training data collection. But in general, choosing codes according to performance on the training set is useful for overall performance on the test set.

