# OpenReview forum: "Evolving Symbolic 3D Visual Grounder with Weakly Supervised Reflection"
_ICLR.cc/2025/Conference — ICLR 2025 Conference Withdrawn Submission_

### Official Review · Reviewer_xFxS · 2024-10-31

**Soundness:** 2
**Presentation:** 1
**Contribution:** 2
**Rating:** 3
**Confidence:** 4

**Summary:**

This work introduces EASE (Evolvable Symbolic Visual Grounder), a fully interpretable symbolic framework for 3D visual grounding. EASE leverages symbolic expressions translated by a large language model (LLM) to compute feature representations for various concepts using explicit Python programs.

**Strengths:**

1. EASE leverages large language models to autonomously generate explicit relation encoders through self-refinement, eliminating the need for human-provided knowledge.

2. EASE surpasses previous training-free methods on the Nr3D benchmark。

**Weaknesses:**

Minor issues:

1. The legend of Figure 1 is too small.

2. The caption of Figure 2 seems not aligned with the diagram.

3. Replace Figure 6 with a high-resolution version.

4. Figure 1 is not referenced.

Major issues:

1. The introduction is very poorly written. (1) the introduction of the significance of the researched problem is missing. (2) The agent-based approaches have not been discussed, which is one of branch of previous work.

2. LLMs are not unnecessarily to every task. It is not advisable to use LLMs for the sake of using LLMs. 50.7% accuracy doesn't looks a good performance. I suggest two possible ways here: (1) You can do a comprehensive analysis to investigate if LLMs are suitable to this task and provide some insights. Or (2) you compare the proposed the method with the state-of-the-art methods to show the effectiveness or efficiency.

3. This work lacks sufficient self-containment and omits essential background information, leading to a structure that compromises readability. Specifically, the discussion of existing work is insufficient, which weakens the persuasiveness of the contribution. For instance, Figure 2 is intended to help readers grasp the narrative, but its impact is limited due to a lack of necessary explanation, making it difficult to understand.

**Questions:**

See "Weakness"

---

### Official Review · Reviewer_P8sa · 2024-11-02

**Soundness:** 3
**Presentation:** 2
**Contribution:** 2
**Rating:** 3
**Confidence:** 3

**Summary:**

This paper introduces a novel application of multimodal PLM, that is, 3D visual grounding. Traditionally, such task require large-scale end-to-end supervised learning to achieve robust accuracy. In this paper, the authors claim a novel framework termed as EASE to adress the training inefficiency of previous supervised systems. Concretely, the authors leverage the powerful reasoning ability and multi-modal information processing ability of GPT-4o model to evolve a scoring functioning program for each relationship in the 3D scene. It only requires a small part of high quality samples to serve as the context adjustment for LLM to get proper program. Then the programs are used as the spatial relation detector to reasoning the correct target object step-by-step. The results in the experimental analysis reveal that EASE achieves comparable grounding perforamance to some baselines, while showing certain gap compared with the SOTA ones.

**Strengths:**

This paper presents a novel application avenue of multimodal PLM-3D visual grounding, which is still at its early stage. The post-processing of the relationship expressions by self-reflective PLM code generation is an interesting idea to automate the grounding techniques. Besides, the experimental results in this paper are comprehensive, objective enough. The ablation studies also provide certain insights.

**Weaknesses:**

Several weaknesses make this paper further the acceptance threshold:

1) The writing, in particular, seems hasty. A systematic proofread is eagerly anticipated for the introduction and methodology part. The content presented in Fig. 1-3 are not well explained in the corresponding paragraph, making it very hard to correlate the figure content with the text description. In particular, in Fig. 2, the authors attempt to make it clear the difference across different methodologies including their own. However, the presentation of their own method (right most) is too messy to tell the novelty of this paper. The following points are some questions caused by the writing, which confuses me a lot.

2) What is the essential advantage of PLM-based grouding method such as EASE? The performance results in this paper is only comparable with SOTA, while the efficiency proof is missing.

3) Further, what is the motivation of using PLM for grounding, can the authors provide any in-depth elaboration?

4) There might be a overclaim in this paper. The authors have claimed that EASE relieve the human expertise for solving grounding task. However, the prompt design presented in Appendix involves nonnegligible expert-level knowledge to make the framework operatable. For example, the helpful tips in line 718-730. An actionable ablation could be removing such hints and observe the performance.

5) Minor: line 087-089, these two lines seems repetitive.

**Questions:**

See weaknesses.

---

### Official Review · Reviewer_7jMC · 2024-11-03

**Soundness:** 2
**Presentation:** 3
**Contribution:** 2
**Rating:** 5
**Confidence:** 3

**Summary:**

This paper introduces ``EASE``, a fully interpretable symbolic framework for 3D visual localization. ``EASE`` leverages Python code generated by LLMs to compute features and employs pretrained classifiers to handle relational and categorical features, enhancing reasoning efficiency and performance.

**Strengths:**

* **Full Interpretability**: ``EASE`` provides a fully interpretable framework.
* **Performance and Efficiency**: ``EASE`` enhances accuracy while reducing inference time and costs.
* **Weak Supervised Learning**: By optimizing the program library with LLMs, ``EASE`` reduces reliance on large-scale training data.

**Weaknesses:**

* **Limitations of Interpretability**: Although ``EASE`` offers a certain level of interpretability, the internal workings of LLMs remain a black box, which may limit the overall interpretability of the framework.
* **Strong engineering nature**：A significant part of the article is exploring how to make LLMs generate better Python code, which may be highly dependent on the prompt and difficult to evaluate from an academic research perspective.

**Questions:**

See ``Weakness``

---

### Official Review · Reviewer_Luiq · 2024-11-03

**Soundness:** 3
**Presentation:** 3
**Contribution:** 3
**Rating:** 6
**Confidence:** 2

**Summary:**

The paper aims at solving the 3D visual grounding problem with less inference cost and superior performance compared with previous llm agent-based method. Given a symbolic expression of a grounding description translated by an LLM, the proposed method, EASE, calculates the feature of each concept utilizing a set of explicit programs in Python learned from a tiny subset of the training data. The program library is learned from the training dataset by a LLM-based optimizer. Extensive experiments have been conducted for analyzing the performance and interpretability of the proposed method.

**Strengths:**

- The idea is interesting and effective
- The paper is well-organized and easy to follow
- The experiments are thorough.

**Weaknesses:**

- I am admittedly not an expert of this domain, and would like to hear my colleague reviewers' opinions.

**Questions:**

None.

---

### Official Review · Reviewer_s56j · 2024-11-06

**Soundness:** 1
**Presentation:** 2
**Contribution:** 1
**Rating:** 3
**Confidence:** 4

**Summary:**

This paper introduces a symbolic framework for 3D visual grounding, aiming for interpretability, reduced inference cost, and superior performance. It uses LLM to translate grounding descriptions and generate executor programs. Experiments on ReferIt3D achieved 50.7% accuracy on Nr3D, outperforming most training-free methods with low inference time and cost.

**Strengths:**

1. The primary strengths of this work lie in its superior performance on ReferIt3D (while not surpassing Transcrib3D) and its low inference cost, thanks to the utilization of LLMs.

2.This work leverages LLMs such as GPT-4o to extract Python programs that represent relations, as opposed to the neural models used in previous neural-symbolic methods. This approach significantly accelerates the process of relation extraction.

**Weaknesses:**

• Over-reliance on Large Language Models (LLMs): This work appears to lean heavily on LLMs for various stages, from description parsing to relation extraction. This dependency raises questions about the originality of the technical contributions, as it is unclear what innovations are introduced beyond the application of existing LLM-based approaches.

• Performance Concerns Relative to Transcrib3D: The work does not convincingly address why its performance is weaker than Transcrib3D, which also leverages LLMs for solving the ReferIt3D task. A comparative analysis of EASE and Transcrib3D is necessary to clarify their differences in approach and provide insight into why EASE does not achieve comparable results.

 • Interpretability Claims: The authors assert that EASE offers enhanced interpretability through program-based relation representation, yet the interpretability is inherently limited by the LLMs it depends on. This approach does not address the challenge of understanding or explaining the inner workings of the LLMs themselves, calling into question the claim of improved interpretability.

• Minor Errors: There are several minor typographical errors in the text, such as "we constraints..." on line 528, which should be corrected for clarity and readability.

**Questions:**

• Clarify Unique Technical Contribution: Please specify the distinct technical innovations introduced in this work, particularly in comparison with existing related approaches. Clearly differentiate how this contribution builds on or diverges from other methodologies to establish its uniqueness and relevance.

• Differentiate EASE from Transcrib3D: Provide a clear analysis of the distinctions between EASE and Transcrib3D, both in terms of approach and implementation. Additionally, address the observed performance gap by comparing key aspects of the two systems, highlighting any factors that might explain EASE's relative performance.

• Logical Reasoning Module Contribution: Outline any contributions made to the logical reasoning component within this work. Additionally, analyze the strengths and limitations of the existing reasoning module to offer a balanced view of its effectiveness and areas where further improvements could be made.

**Details Of Ethics Concerns:**

No ethics concerns.

---

### Official Review · Reviewer_fdq8 · 2024-11-08

**Soundness:** 3
**Presentation:** 3
**Contribution:** 3
**Rating:** 5
**Confidence:** 3

**Summary:**

This paper introduces EASE, a novel approach to 3D visual grounding  that uses a symbolic framework supported by large language models. EASE generates interpretable relation encoders using Python programs learned from a subset of training data and continuously refined through LLM-based optimization. The method is designed to address the computational inefficiencies and opacity of traditional neuro-symbolic and end-to-end LLM-based methods. EASE demonstrates competitive performance on the ReferIt3D dataset, with significant improvements in efficiency compared to baseline models.

**Strengths:**

- The symbolic approach offers reduced inference time and token costs compared to agent-based LLM models, making it more practical for large-scale applications.

- The framework eliminates the need for human-provided relation annotations by generating code with LLMs, enhancing scalability and reducing manual effort.

- The use of a unit test-based feedback loop for self-refinement is innovative, ensuring that generated code improves iteratively.

- The experiments include different data splits (e.g., easy vs. hard, view-dependent vs. view-independent), showcasing the model's versatility.

**Weaknesses:**

- While the symbolic framework is interpretable, handling complex relations or compound instructions may pose limitations without significant modifications.

- Although human annotations are not required, the framework still depends heavily on the capabilities of LLMs for generating initial code, which may introduce variability in output quality.

- The system represents scenes as a list of 3D bounding boxes, omitting detailed attributes like object orientation, texture, or color, which could be vital for more nuanced tasks.

- The feedback and code refinement process may not fully resolve deeper logical errors or misinterpretations made by the LLM in code generation.

**Questions:**

- Are there any plans to incorporate object orientation and visual attributes into the scene representation to improve grounding accuracy?

- How adaptable is the framework when extending to different types of 3D visual grounding tasks outside the ReferIt3D dataset?

- How does the performance of EASE compare when using different LLMs for code generation (e.g., GPT-4 versus other models)?

- Can the iterative code refinement approach be parallelized to reduce overall computation time?

- Could additional strategies, such as reinforcement learning, be incorporated to further optimize the LLM-generated code for relation encoding?

---

### Note · Authors · 2024-11-15

I have read and agree with the venue's withdrawal policy on behalf of myself and my co-authors.